# HMGCR inhibition stabilizes the glycolytic enzyme PKM2 to support the growth of renal cell carcinoma

**Jiajun Huang◉*[◎], Xiaoyu Zhao[◎], Xiang Li, Jiwei Peng, Weihao Yang, Shengli Mi***

Bio-manufacturing Engineering Laboratory, Tsinghua Shenzhen International Graduate School, Tsinghua University, Shenzhen, Guangdong, China

◎ These authors contributed equally to this work.
* vt_bongo@hotmail.com (JH); mi.shengli@sz.tsinghua.edu.cn (SM)

**Data Availability Statement:** All relevant data are within the paper and its Supporting Information files.

**Funding:** This work was funded by the Project of Basic Research of Shenzhen, China

## Abstract

Renal cell carcinoma (RCC) is responsible for most cases of the kidney cancer. Previous research showed that low serum levels of cholesterol level positively correlate with poorer RCC-specific survival outcomes. However, the underlying mechanisms and functional significance of the role of cholesterol in the development of RCC remain obscure. 3-Hydroxy-3-methylglutaryl coenzyme A reductase (HMGCR) plays a pivotal role in RCC development as it is the key rate-limiting enzyme of the cholesterol biosynthetic pathway. In this study, we demonstrated that the inhibition of HMGCR could accelerate the development of RCC tumors by lactate accumulation and angiogenesis in animal models. We identified that the inhibition of HMGCR led to an increase in glycolysis via the regulated HSP90 expression levels, thus maintaining the levels of a glycolysis rate-limiting enzyme, pyruvate kinase M2 (PKM2). Based on these findings, we reversed the HMGCR inhibition-induced tumor growth acceleration in RCC xenograft mice by suppressing glycolysis. Furthermore, the coadministration of Shikonin, a potent PKM2 inhibitor, reverted the tumor development induced by the HMGCR signaling pathway.

## Introduction

Renal cancer is one of the top 10 cancers in terms of mortality and continues to increase in terms of prevalence [1]. Renal cell carcinoma (RCC) accounts for the majority of adult kidney cancers. The most effective clinical treatments for RCC are surgical resection, targeted therapy against vascular endothelial growth factor (VEGF), and immunotherapy agents such as novel kinase inhibitors [2]. Unfortunately, approximately 20% to 30% of patients with RCC will develop metastatic renal cell carcinoma (mRCC) after clinical diagnosis. Furthermore, approximately 30% of newly diagnosed patients with localized disease experience metastases [3]. Due to tumor recurrence and metastasis, the clinical outcomes of patients with RCC have not shown satisfactory improvement over recent years [4].

As a metabolic disease, RCC is associated with multiple preoperative surrogate markers; of these, cholesterol is a promising prognostic candidate for clinical treatment. Although

(JCYJ20180507183655307 & JCYJ20190813143221901). SL Mi and JJ Huang received the funding. The funders had no role in study design, data collection and analysis, decision to publish, or preparation of the manuscript.

**Competing interests:** The authors have declared that no competing interests exist.

**Abbreviations:** AKG, α-ketoglutarate; β-cat, β-catenin; BCA, bicinchoninic acid; CAF, cancer-associated fibroblast; CHX, cycloheximide; Co-IP, co-immunoprecipitation; CoQ10, coenzyme Q10; ECAR, extracellular acidification rate; EGFR, epidermal growth factor; eNOS, endothelial NO synthase; FBS, fetal bovine serum; FCCP, carbonyl cyanide 4-(trifluoromethoxy) phenylhydrazone; FPP, farnesyl pyrophosphate; GGPP, geranylgeranyl pyrophosphate; GSH, glutathione; GST, glutathione S transferase; HIF-1α, hypoxia-inducible factor-1α; HMGCR, 3-hydroxy-3-methylglutaryl coenzyme A reductase; HSF-1, heat shock transcription factor-1; HSP, heat shock protein; HSP90, heat shock protein 90; HSR, heat shock response; mRCC, metastatic renal cell carcinoma; MSC, mesenchymal stem cell; MVA, mevalonate; NF-κB, nuclear factor-κB; NO, nitric oxide; OCR, oxygen consumption rate; PEP, phosphoenolpyruvate; PKM2, pyruvate kinase M2; RCC, renal cell carcinoma; *sh*RNA, short hairpin RNA; SRB, sulforhodamine B; STAT3, signal transductors, transcriptional activators 3; TCA, trichloroacetic acid; TIF, translation initiation factor; TME, tumor microenvironment; TNF-α, tumor necrosis factor-α; VEGF, vascular endothelial growth factor.

conflicting data exists for the relationship between cholesterol and the risk of cancer, an increasing body of experimental evidence now indicates that disorders of cholesterol metabolism do contribute to lymphatic metastasis and a poor prognosis in RCC [5,6]. A large multicenter cohort study involving 3,046 RCC patients, suggested that low preoperative serum cholesterol levels were closely related to more aggressive tumor features and worse cancer-specific survival outcomes [7]. In addition, a 1:2-matched case–control study reported that elevated serum levels of cholesterol, LDL cholesterol, and HDL cholesterol were linked with a reduced risk of RCC [8]. However, the underlying mechanism that associates cholesterol metabolism with RCC remains unclear.

The mevalonate pathway (MVA pathway) is the most important metabolic pathway by which mammals synthesize cholesterol and isoprenoid products. 3-Hydroxy-3-methylglutaryl coenzyme A reductase (HMGCR) is the main rate-limiting enzyme in the MVA pathway. This enzyme controls more than 20 subsequent enzymatic reactions until the final product (cholesterol) is synthesized [9]. Recently, a number of studies involving in vitro or in vivo trials have revealed that the down-regulation of HMGCR correlates with various tumor progression and cancer survival [10–13]. It is also worth noting that the statistical analysis of the TCGA database showed that the expression levels of HMGCR mRNA in RCC cells were significantly reduced compared to nonmalignant renal cells [10]. Collectively, the available data imply that the development of RCC is related to inhibition of the cholesterol synthesis pathway and that HMGCR plays an important role in the development of tumors by an underlying mechanism that has yet to be elucidated.

Glycolysis is an intracellular metabolic process that consists of 10 consecutive enzymatic reactions that degrade 1 molecule of glucose into 2 molecules of pyruvate. Increased levels of aerobic glycolysis is a characteristic of rapid cell proliferation [14]. Enhanced glycolysis has important significance for tumors for several reasons: because it provides the energy required for the rapid growth of tumor cells; because the intermediate products of glycolysis are important precursors for cell biosynthesis; because increased glycolysis is conducive to maintaining the reducing power of the cell; because it promotes tumor angiogenesis; and because it activates other oncogenic factors, which directly leads to tumor proliferation. Existing research shows that inhibition of glycolysis leads to disorders of cholesterol synthesis [15]. However, studies relating to the cholesterol synthesis pathway and how this pathway can regulate glycolysis are still rare. One recent study indicated that tumors with higher expression levels of glycolytic genes, but lower levels of genes related to the MVA pathway, are likely to be more aggressive than tumors with a more cholesterogenic phenotype [16]. In addition, the response of the tumor microenvironment (TME) to changes in cellular glycolysis becomes an external signal that can then affect cholesterol metabolism [17]. Despite such research, the underlying mechanism responsible for how the MVA pathway regulates glycolysis during tumor development remains unclear.

In the present study, we demonstrated that the inhibition of HMGCR, a rate-limiting enzyme in the MVA pathway, was associated with increased lactate production and the promotion of RCC growth. We found that the inhibition of HMGCR enhanced cellular glycolysis by increasing the protein concentration of PKM2 and that HSP90 played a key role in reducing PKM2 protein degradation during this process. The administration of Shikonin, a drug that specifically inhibits PKM2, reduced the levels of glycolysis and suppressed tumor growth caused by the HMGCR inhibitor. In summary, our research revealed that HMGCR inhibition induced the enhancement of glycolysis in tumors, thus suggesting that this is the mechanism responsible for reduced cholesterol synthesis during the development of RCC. We also provided a feasible therapy for the development of a feasible therapy for RCC patients that would involve a medication that targets glycolysis.

## Results

### HMGCR inhibition led to an acceleration of RCC tumor growth

The primary purpose of this study was to determine the effect of HMGCR inhibition on the in vivo growth of tumors derived from human renal cancer cells. We first established a xenograft model of human renal adenocarcinoma cell line ACHN in nude mice and induced HMGCR inhibition with lovastatin; the negative control group was given normal saline. Mice in each group developed palpable tumors, and the accelerated tumor growth was measured and recorded at 3- to 4-day intervals (Fig 1A). We found that the lovastatin-treated group had a significantly lower serum level of cholesterol (Fig 1C) but greater tumor size compared to the control group (Fig 1A). Tumors in the lovastatin-treated group also weighed more than those in the control group at necropsy (Fig 1B). By examining the resected tumor tissue, we found that the lactate content in the tumor tissue of the lovastatin-treated group was significantly higher by 25% ± 8.1% ($p = 0.006$) (Fig 1D); the levels of VEGF and glutathione (GSH) in the tissues were also significantly increased (Fig 1E and 1F). Moreover, tumor neovascularization testing (Fig 1G) and CD31 staining (Fig 1H) indicated that the angiogenic response in the lovastatin treatment group was significantly stronger than that in the saline group. These results suggested that treatment with HMGCR inhibitors had made the microenvironment of the tumor more conducive to growth. It has been established that elevated lactate levels in tumors often precede increased levels of glycolysis in tumor cells. Glycolysis provides energy and materials for the rapid proliferation of tumor cells and creates an acidic and inflammatory TME, thereby promoting tumor growth and angiogenesis. Moreover, the accelerated growth of tumors promoted acidification and hypoxia in the TME. Based on this inference, we preliminary hypothesized that HMGCR inhibition promoted tumor growth by increasing glycolysis.

### HMGCR inhibition did not directly promote the proliferation of tumor cells but enhanced glycolysis

To clarify that the accelerated growth of tumors caused by HMGCR inhibition was due to the enhanced glycolysis of tumor cells, we cultured ACHN and 786-O cells in standard growth medium with lovastatin in a range of concentrations. However, the measurement of cell numbers indicated that lovastatin only led to negligible influence on cell proliferation (S1 Fig). This finding was corroborated by the treatment of ACHN and 786-O cells with other statins at doses of 1 μM; this also caused negligible effects on cell proliferation (S1 Fig). The results of this assay implied that HMGCR inhibition did not correlate with tumor growth via an effect on cell proliferation. Next, we measured lactate produced by ACHN and 786-O cell lines after lovastatin stimulation. At the indicated concentrations, lovastatin augmented the release of lactate into the culture medium in ACHN cells by 24% to 69% and in 786-O cells by 22% to 55% (Fig 2A). To confirm the specificity of lactate production enhanced by lovastatin-inhibited HMGCR, we established ACHN and 786-O cells stably expressing short hairpin RNA (*sh*RNA) targeting HMGCR (S1 Fig). Consistent with the previous results, the extracellular and intracellular levels of lactate both showed elevation in HMGCR knockdown cells (Fig 2B and 2C). The extracellular acidification rate (ECAR) at both baseline and maximum level (when stimulated by the mitochondrial ATP synthase inhibitor oligomycin) increased with HMGCR inhibition or knockdown (Fig 2D, S1 Fig). Furthermore, supplementation with MVA, a downstream product of HMGCR, restored lovastatin-induced lactate alterations in ACHN cells (Fig 2E). In addition, increased lactate production was accompanied by increased consumption of glucose. This was confirmed by the direct measurement of residual glucose in

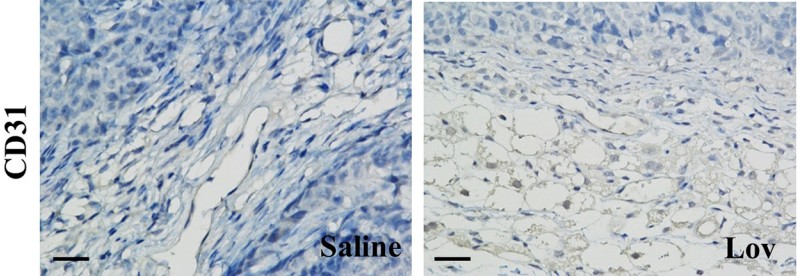

**Fig 1. Inhibition of HMGCR promoted RCC tumor growth.** (A) Tumor diameters of lovastatin-treated xenograft mice. ACHN cells were injected into bilateral flank of mice. Healthy control group were treated with saline and lovastatin-treated group were oral administered with lovastatin (Lov) at a dose of 5 mg/kg 2 days after modeling. (B) Xenograft tumors were resected and weighed after mice were killed on day 49. (C) Serum cholesterol levels on day 49 ($n = 10$ each group) after last treatment. (D) Lactate, (E) VEGF, and (F) GSH level in resected tumor. (G) Xenografts neovascularization were measured by the Drabkin's reagent kit. (H) Immunohistochemical staining of CD31 in ACHN xenografts (scale bar: 50 μm). Data are shown as mean ± SD, $^*(p \leq 0.05)$, $^{**}(p \leq 0.01)$, or $^{***}(p \leq 0.001)$. GSH, glutathione; HMGCR, 3-hydroxy-3-methylglutaryl coenzyme A reductase; Lov, lovastatin; RCC, renal cell carcinoma; VEGF, vascular endothelial growth factor.

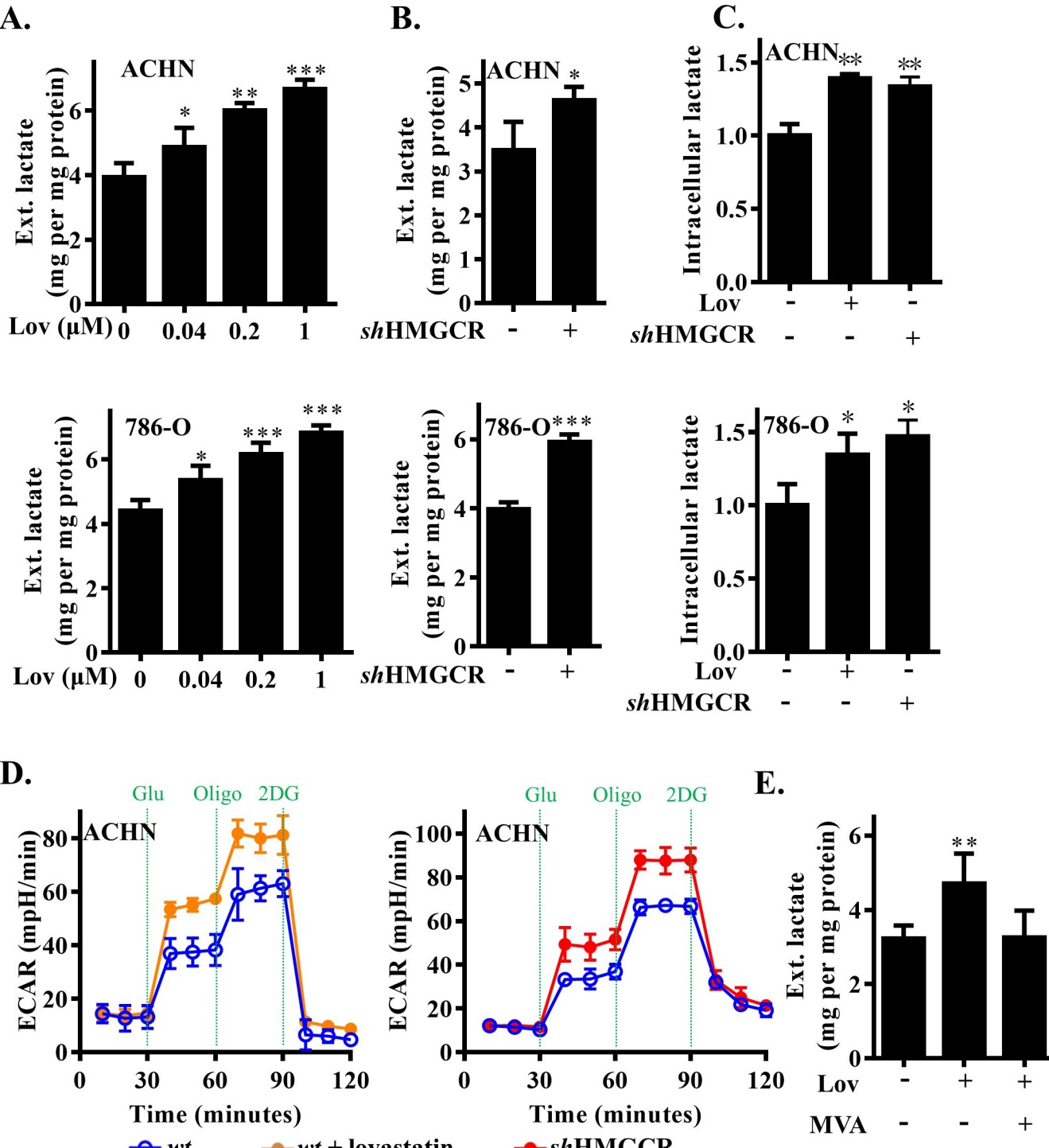

**Fig 2. HMGCR inhibition enhanced RCC cells glycolytic.** (A) Dose-dependent analysis of extracellular lactate level in ACHN and 786-O cells treated with lovastatin. (B) Extracellular lactate level in HMGCR knockdown and wild-type cells. (C) Intracellular lactate level in lovastatin, HMGCR knockdown, or wild-type RCC cells. (D) Kinetic ECAR response of *sh*HMGCR-ACHN cells or lovastatin-stimulated ACHN cells to 10 mM glucose, 2 μM oligomycin, and 100 mM 2-DG. (E) Extracellular lactate level in ACHN cells with or without lovastatin and mevalonate treatment. The data are represented as the mean ± SD from 3 independent experiments, $^*(p \leq 0.05)$, $^{**}(p \leq 0.01)$, or $^{***}(p \leq 0.001)$. 2DG, 2-deoxy-D-glucose; ECAR, extracellular acidification rate; Glu, glucose; HMGCR, 3-hydroxy-3-methylglutaryl coenzyme A reductase; Lov, lovastatin; MVA, mevalonate; Oligo, oligomycin; RCC, renal cell carcinoma; *wt*, wild type.

the culture medium and flow cytometry analysis by the fluorescent deoxyglucose analog 2-NBDG (S1 Fig).

## PKM2 protein contributed to the elevation of glycolysis induced by the inhibition of HMGCR

Lactate is the end product of anaerobic glycolysis. Therefore, we reasoned that the observation of an increasing level of lactate in RCC cells was likely to be attributed to change in the functionality of glycolytic enzymes. To identify the potential mechanism involved, we examined the glycolysis-related regulators expression level under HMGCR inhibition, including Hif-1α, Akt, Tigar, c-myc, Glut1, HK1/2, Enolase1, LDHA, and PKM2 (S2 Fig) in ACHN cells. Indeed, we found an abnormal increase in the levels of PKM2 protein in the tumor tissue of lovastatin-treated mice (Fig 3A). The expression of PKM2 on the tumor cells of our in vitro culture model increased in response to lovastatin treatment; this occurred in both a dose- and time-dependent manner (Fig 3B). These results were also reproduced upon treatment with other statins (Fig 3C). Furthermore, existing reports prove that PKM2 not only acts as a rate-limiting enzyme to control the rate of glycolysis in tumor cells; it also regulates the secretion of VEGF and promotes tumor angiogenesis [18]. Therefore, we turned our attention to PKM2. The specificity of the PKM2 regulation following the inhibition of HMGCR by statins was confirmed by both HMGCR knockdown and MVA supplementation experiments in ACHN cells (Fig 3D and 3E).

To verify that HMGCR inhibition specifically regulated cell glycolysis via PKM2, we used lentivirus-mediated shRNA to knock down PKM2 in ACHN cells (S2 Fig). We found that a reduction in the expression of PKM2 protein level resulted in a decline in lactate production and glucose consumption; this effect could not be reversed by the further application of lovastatin (Fig 3F, S2 Fig). In contrast, the exogenous expression of PKM2 (S2 Fig) significantly increased lactate production and glucose consumption in ACHN cells (Fig 3G, S2 Fig). In addition, the level of VEGF was also augmented with the exogenous expression of PKM2 (Fig 3H); this was consistent with our findings from the animal model. To understand the mechanisms underlying the HMGCR inhibition-induced PKM2 up-regulation, we first measured the mRNA levels of PKM2 in ACHN cells. There was no significant change either with lovastatin treatment or with HMGCR knockdown (S3 Fig). This suggested that the inhibition of HMGCR might affect the stability of the PKM2 protein. Furthermore, the enzymic activity of PKM2 was inversely related to the regulation of HMGCR (Fig 3I).

Cycloheximide (CHX) is an inhibitor of protein synthesis and specifically inhibits the synthesis of proteins in the cytoplasm. Next, we performed CHX-chase assays to evaluate the stability of the PKM2 protein. As shown in Fig 3J, when treated with lovastatin ACHN cells clearly increased the half-life of the PKM2 protein. Correspondingly, mevalonate supplementation substantially restored the increasing half-life of the PKM protein that was induced by lovastatin in ACHN cells (Fig 3J). Moreover, a proteasome inhibitor (MG132) rescued the increase of PKM2 induced by lovastatin and shHMGCR interference (Fig 3K). These data revealed that HMGCR inhibition increased the levels of PKM2 protein by slowing down degradation by the proteasome and by promoting the stability of PKM2.

## HSP90 is a critical regulator of PKM2 protein augmentation

Heat shock protein 90 (HSP90) is an oncoprotein that regulates protein conformation. A previous report stated that the protein levels of PKM2 were regulated by HSP90 [19]. Therefore, we examined the protein level of HSP90 in ACHN cells after the inhibition of HMGCR. As expected, lovastatin increased both HSP90 and PKM2 concentrations in a dose-dependent

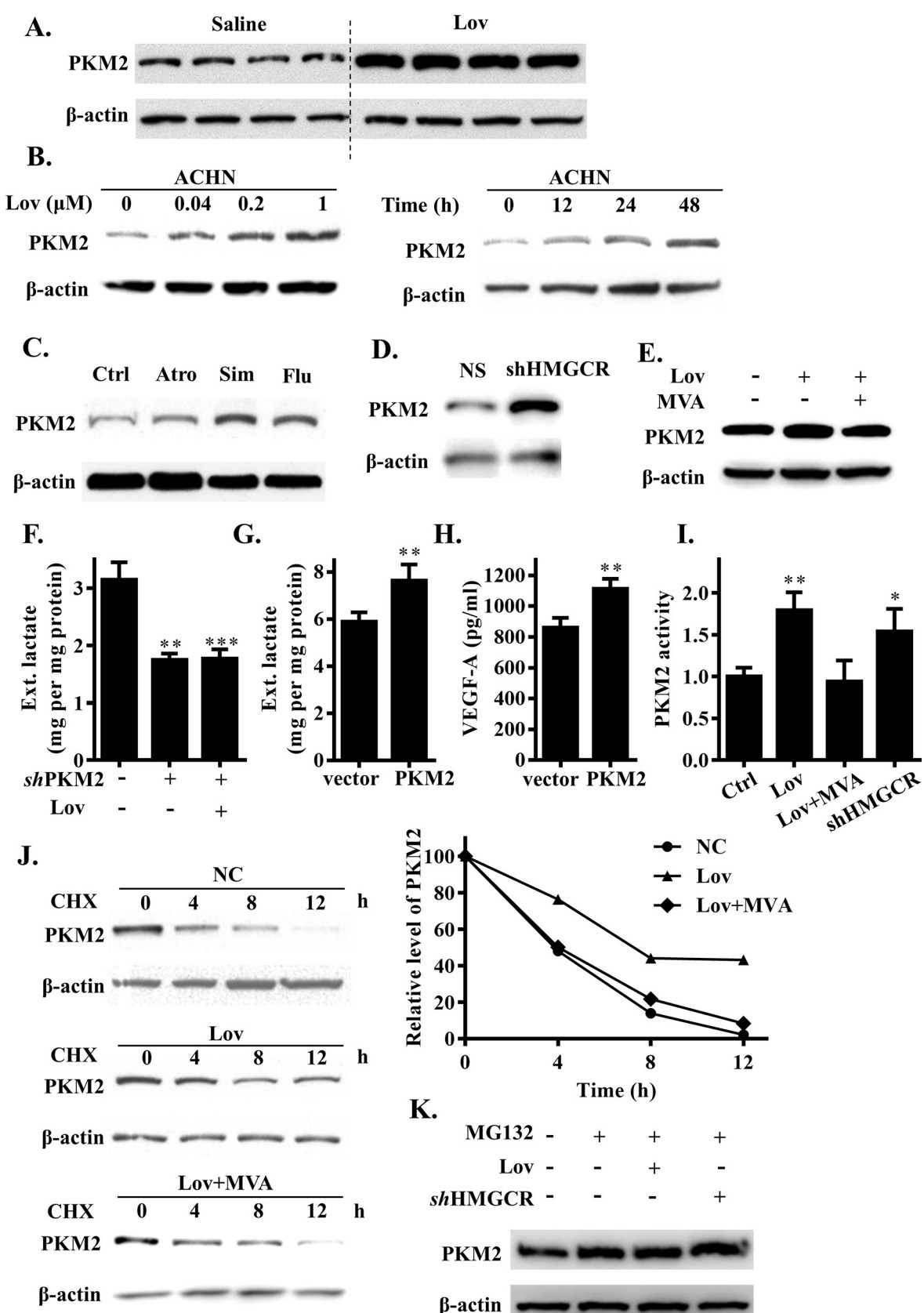

**Fig 3. HMGCR inhibition enhanced PKM2 stability.** (A) PKM2 protein levels in xenograft mice tumor tissue. (B) Dose- and time-dependent analysis of PKM2 expression level in lovastatin-treated ACHN cells. (C) PKM2 protein expression level in ACHN cells treated with 1 μM atorvastatin (Atr), simvastatin (Sim), or fluvastatin (Flu) for 24 hours. (D) PKM2 protein expression level in HMGCR knockdown or wild-type ACHN cells. (E) PKM2 protein expression level in ACHN cells with or without lovastatin and mevalonate treatment. (F) Extracellular lactate level in cells of lovastatin treatment or PKM2 shRNA interference. (G) Extracellular lactate level in PKM2 overexpression ACHN cells. (H) Extracellular VEGF level in PKM2 wild-type and overexpression ACHN cells. (I) PKM2 protein activity level in HMGCR inhibition ACHN cells. (J) ACHN cells were subject to lovastatin or together with mevalonate treatment; PKM2 protein half-life in ACHN cells was analyzed following CHX stimulate. (K) PKM2 protein level in ACHN cells after inhibiting proteasomal degradation by MG-132. Data are shown as mean ± SD, $^*(p \leq 0.05)$, $^{**}(p \leq 0.01)$, or $^{***}(p \leq 0.001)$. Atr, atorvastatin; CHX, cycloheximide; Ctrl, control; Flu, fluvastatin; HMGCR, 3-hydroxy-3-methylglutaryl coenzyme A reductase; Lov, lovastatin; MVA, mevalonate; NC, negative control; NS, non-specific; PKM2, pyruvate kinase M2; shRNA, short hairpin RNA; Sim, simvastatin; VEGF, vascular endothelial growth factor.

manner (Fig 4A). Then, we attenuated the expression of HSP90 protein using shRNA (S4 Fig) and found that the effect of lovastatin on increasing PKM2 had been eliminated (Fig 4B). Moreover, HSP90 knockdown directly reduced levels of PKM2 (Fig 4B). These findings suggested that the inhibition of HMGCR might regulate PKM2 via HSP90. To verify this observation, we performed co-immunoprecipitation (Co-IP) analysis. As shown in Fig 4C, lovastatin led to the increasing recruitment of HSP90 to PKM2 (Fig 4C). Notably, the knockdown of HSP90 by shRNA restored the enhanced PKM2 enzyme activity and lactate production caused by lovastatin (Fig 4D and 4E). In addition, RT-PCR data showed that lovastatin treatment and HMGCR knockdown increased the mRNA levels of HSP90 in ACHN cells (Fig 4F). Following the supplementation of various metabolites downstream of HMGCR, the rise in HSP90 protein concentration appeared to be attenuated to varying degrees (S4 Fig). Furthermore, we found that the inhibition of HMGCR led to an increase in the level of heat shock transcription factor-1 (HSF-1); this may play represent a possible pathway for the regulation of HSP90 (S4 Fig). Collectively, these data indicate that HSP90 plays a critical role in the regulation of PKM2 in cells when inhibited by HMGCR.

## HSP90 increased the abundance of PKM2 by inhibiting its degradation by the proteasome

To characterize the mechanisms governing the manner by which HSP90 modulates PKM2 abundance, we first performed reciprocal Co-IP and glutathione S transferase (GST) pull-down assays and found that HSP90 could directly interact with PKM2 (Fig 5A and 5B). Then, we investigated whether the interaction between PKM2 and HSP90 would affect the abundance of PKM2 in ACHN cells. HSP90 shRNA was transfected into ACHN cells and led to a reduction in PKM2 protein level with reduced HSP90 expression (Fig 5C). In contrast, the exogenous expression of HSP90 significantly increased the abundance of PKM2 in ACHN cells (Fig 5C). Next, we induced protein degradation in ACHN cells via CHX and found that the half-life of PKM2 changed when HSP90 was either knocked out or overexpressed (Fig 5D, S4 Fig). Simultaneously, MG132 restored the protein levels of PKM2 that had been reduced in response to HSP90 knockdown (Fig 5E). Furthermore, we performed the ubiquitination assay for wild-type PKM2. Consistent with its degradation by the proteasomal pathway, PKM2 was extensively ubiquitinated in ACHN cells in which HSP90 had been knocked down (Fig 5F). Collectively, these data indicated that HSP90 was critical for reducing the degradation and maintaining the stability of PKM2.

## The suppression of glycolysis abolished HMGCR inhibition-induced tumor growth in vivo

Based on above data, we speculated that tumor growth in response to HMGCR inhibition in mice would be eliminated if glycolysis was suppressed. To test this hypothesis, we used

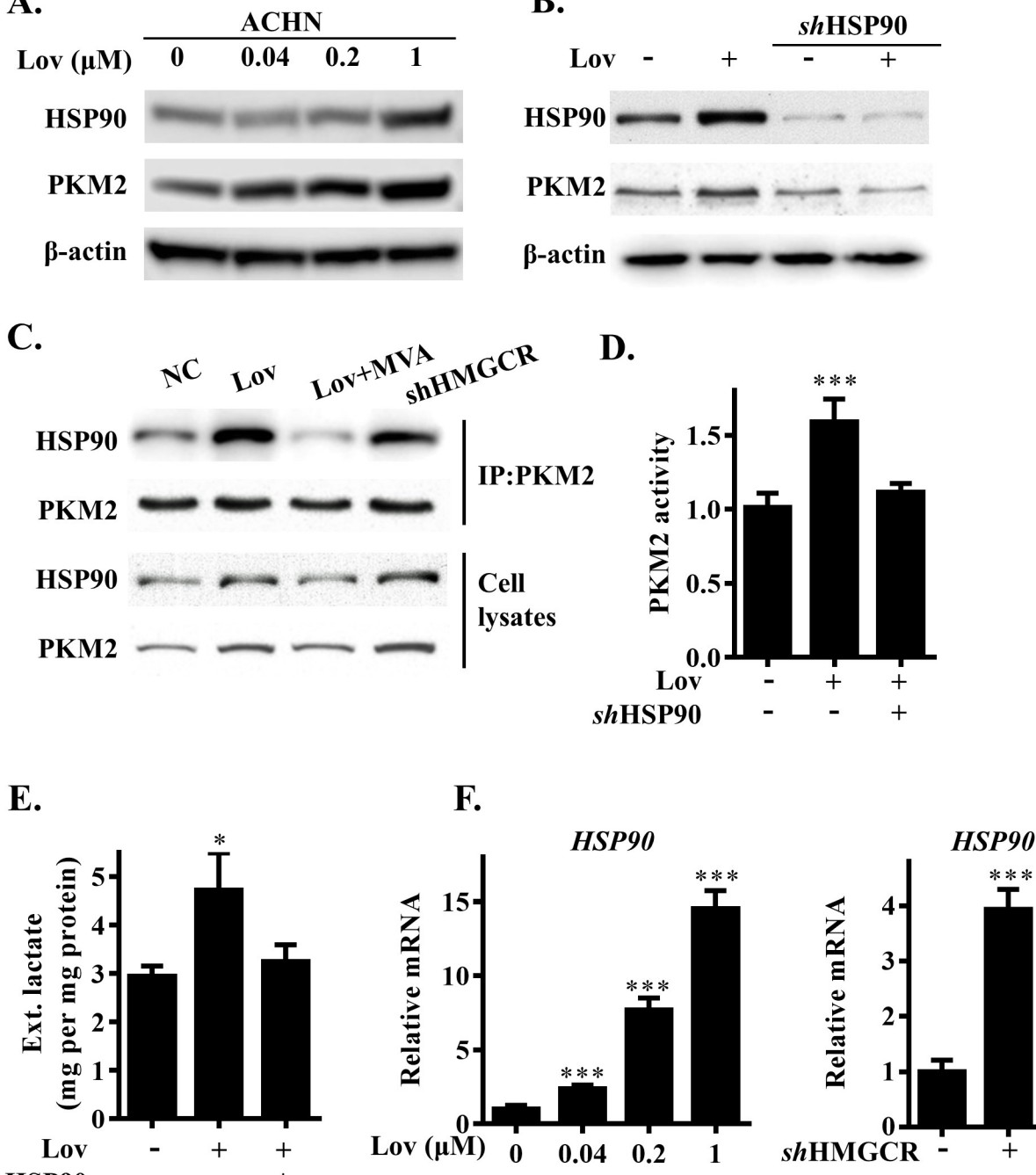

**Fig 4. HMGCR inhibition increased PKM2 protein level by up-regulating HSP90 expression.** (A) Dose-dependent analysis of PKM2 and HSP90 level in lovastatin-treated ACHN cells. (B) PKM2 level in wild-type or HSP90 knockdown ACHN cells with lovastatin treatment. (C) Immunoprecipitation assay was performed using PKM2 antibody in indicated treatment ACHN cells. (D) PKM2 protein activity level in wild-type or HSP90 knockdown ACHN cells with lovastatin treatment. (E) Extracellular lactate level in wild-type or HSP90 knockdown ACHN cells with lovastatin treatment. (F) Dose-dependent analysis of HSP90 expression levels in lovastatin-treated ACHN cells (left panel). HSP90 expression levels in HMGCR knockdown ACHN cells (right panel). The data are represented as the mean ± SD from 3 independent experiments, $^*$($p \leq 0.05$), $^{**}$($p \leq 0.01$) or $^{***}$($p \leq 0.001$). HMGCR, 3-hydroxy-3-methylglutaryl coenzyme A reductase; HSP90, heat shock protein 90; IP, immunoprecipitation; Lov, lovastatin; MVA, mevalonate; NC, negative control; PKM2, pyruvate kinase M2.

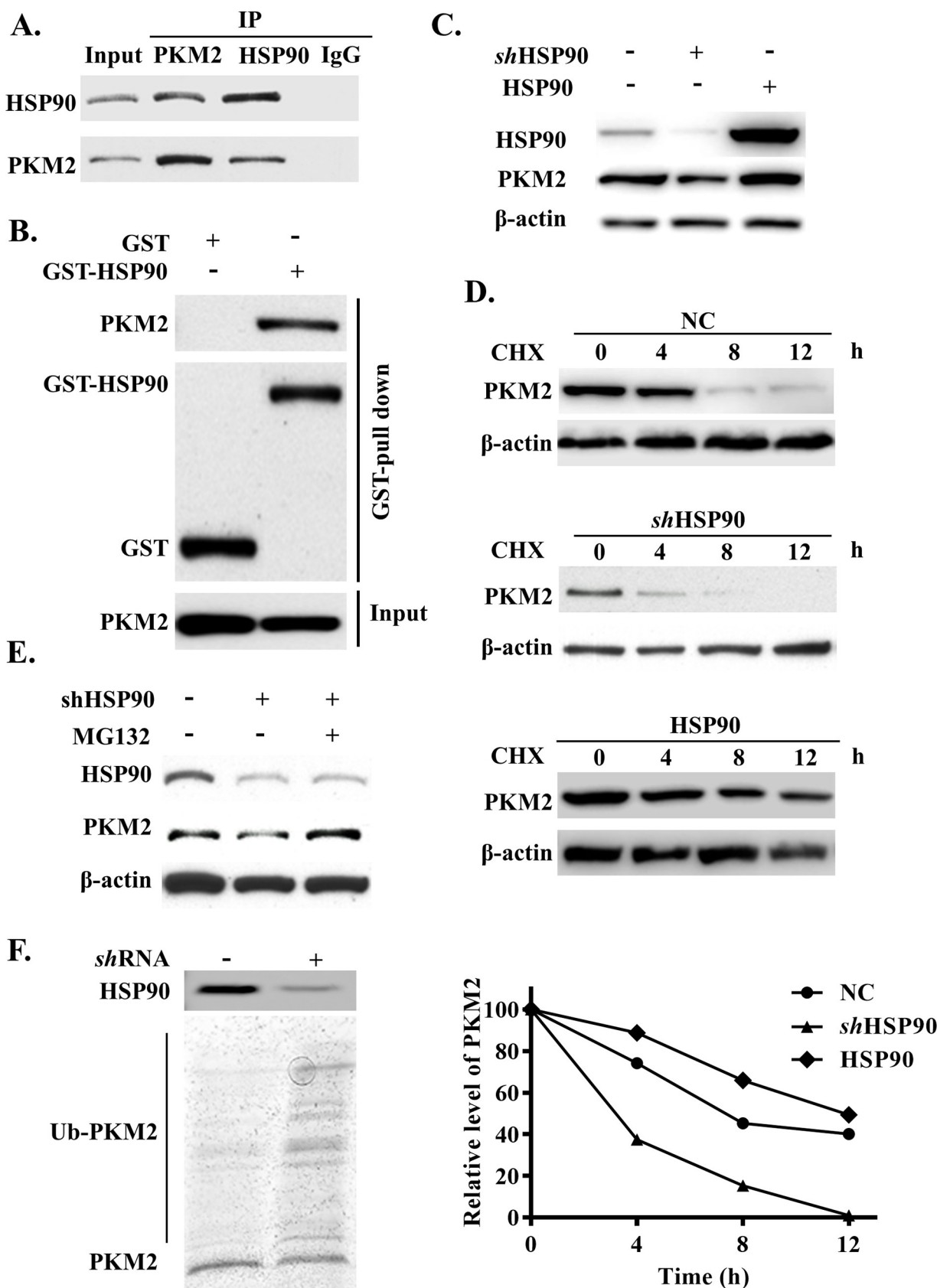

**Fig 5. HSP90 promoted the PKM2 protein stability in ACHN cells.** (A) Endogenous HSP90 and PKM2 proteins interaction was analysis by Co-IP and (B) GST pull-down assays. (C) PKM2 protein were analysis in HSP90 knockdown and overexpression ACHN cells. (D) PKM2 protein half-life was analysis in CHX stimulate HSP90 knockdown and overexpressed ACHN cells. (E) PKM2 levels were analyzed in HSP90 knockdown ACHN cells with MG132 treatment. (F) Ubiquitination of PKM2 in HSP90 knockdown ACHN cells. The data are represented as the mean ± SD from 3 independent experiments. CHX, cycloheximide; Co-IP, co-immunoprecipitation; GST, glutathione S transferase; HSP90, heat shock protein 90; IgG, immunoglobulin G; IP, immunoprecipitation; NC, negative control; PKM2, pyruvate kinase M2; *sh*RNA, short hairpin RNA.

xenograft mice transplanted with *sh*PKM2 cells to observe tumor growth after HMGCR inhibition. In parallel, in the other 2 groups of mice, we administered Shikonin, a PKM2 inhibitor, either with or without lovastatin for 42 days. Shikonin is a major component of a Chinese herbal medicine named zicao (purple gromwell, the dried root of *Lithospermum erythrorhizon*). Shikonin is also able to reduce glycolysis via its actions on PKM2 [20]. As we expected, the growth of tumors in the *sh*PKM2 group and Shikonin intervention group did not show acceleration with the application of lovastatin or *sh*HMGCR (Fig 6A and 6B, S5 Fig). Notably, the PKM2 inhibitor had a better antitumor effect in the HMGCR inhibition model than in the wild type model (Fig 6A and 6B, S5 Fig). The changes in tumor growth were consistent with lactate levels in the tumor tissue, the release of VEGF and GSH, and the restoration of angiogenesis (Fig 6C–6F). In vitro assays, culture medium testing, and cell ECAR monitoring results all showed that the supplementation of Shikonin eliminated the increased lactate production and glucose consumption by lovastatin, to a level that was even lower than the control; these findings were consistent with the potent ability of Shikonin to inhibit glycolysis (S5 Fig). These results proved that the suppression of glycolysis reversed the acceleration of tumor growth caused by HMGCR inhibition and provided antitumor effects in mice.

## Discussion

Conventionally, most investigators have previously considered that a high serum cholesterol level represents a potential risk factor for aggressive cancer progression and a poor prognosis. However, the effects of cholesterol levels on various cancers are heterogeneous. Recent clinical and experimental research has suggested that high levels of serum cholesterol were not associated with the risk of cancer [21,22], while low levels of serum cholesterol were positively correlated with the incidence, aggressiveness, and prognosis of certain cancers including RCC [7,23,24]. One possible explanation for these clinical observations is that tumor-associated hypocholesterolemia may be due to increased cholesterol absorption by tumor cells [25]. However, this could not explain the decline of HMGCR in RCC cells [10]. Another study suggested that cholesterol promoted the translocation of CD44 into lipid rafts and attenuated CD44-Ezrin binding, a process that is essential for cell migration and cancer metastasis [26]. This might be the reason for the good prognosis of aggressive tumors after serum cholesterol reduction had returned to relatively high levels. However, this possibility requires further investigation. In this study, we found that the growth of RCC was related to the strengthening of glycolysis; this provided a new perspective for previous clinical observations.

In this study, we specifically explored the relationship between HMGCR inhibition and RCC formation. We used statins to inhibit HMGCR in a mouse model. Unlike the clinical application of statins in patients with hypercholesterolemia, we used animal models with normal cholesterol levels for research; we considered that this was a more suitable model for the actual situation of RCC patients without a background of hypercholesterolemia. Therefore, we could investigate the evolution of tumors after HMGCR had been directly inhibited; this eliminated the side effects of hypercholesterolemia on RCC. Statins are cholesterol-lowering drugs that have attracted much attention because of their potential anticancer effect by inhibiting the

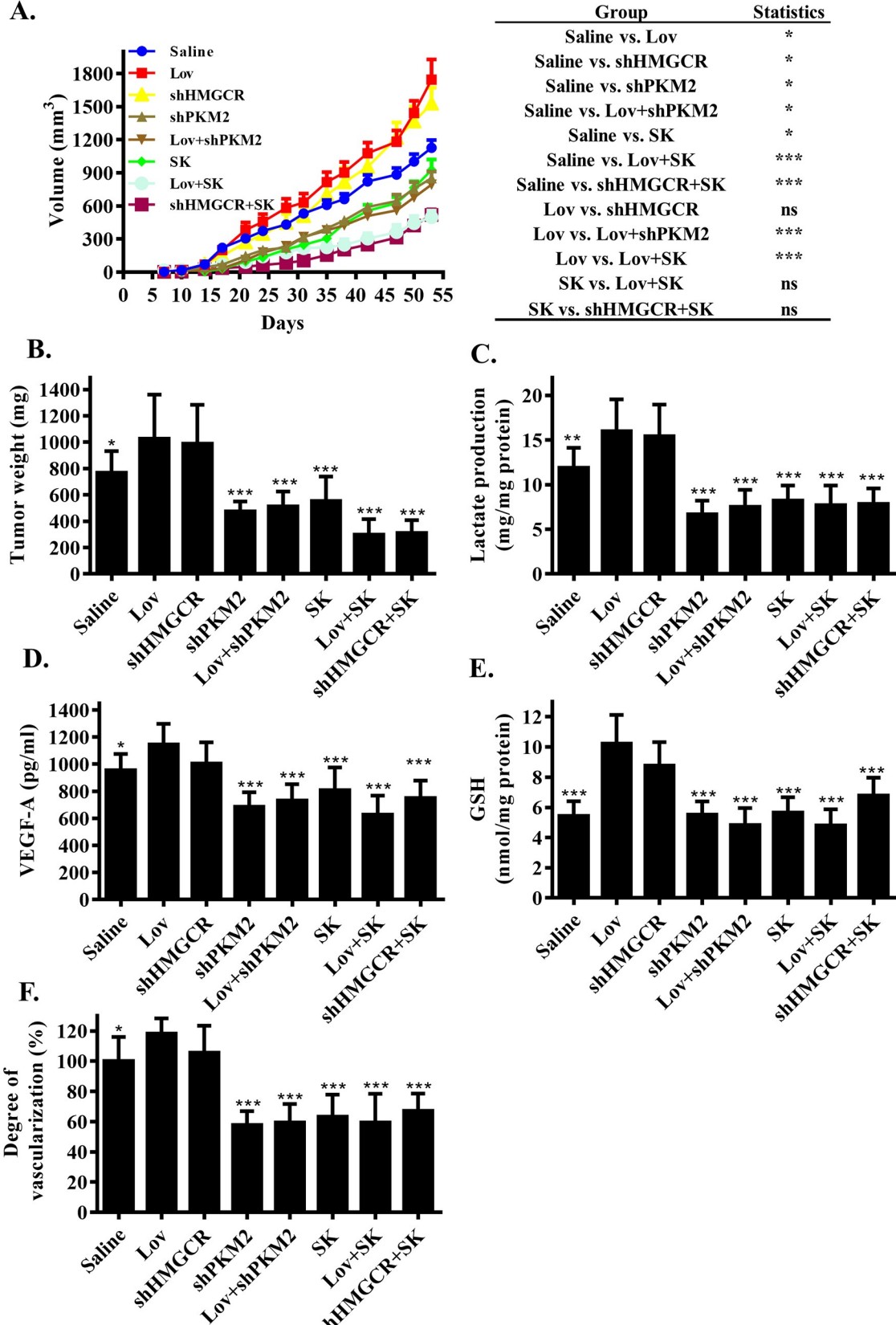

**Fig 6. Glycolysis suppression reverses HMGCR inhibition induced tumor growth in xenograft mice.** (A) Progression of tumor volumes during days 7 to 53 in xenograft mice. Shikonin (SK) treated with 2 mg/kg/2 days; lovastatin treated with 5 mg/kg/2 days. SHCOO2 vector was used as negative control. ($n$ = 10 per group) (B) Xenograft tumors were resected and weighed after mice were killed on day 53. ($n$ = 10 per group) (C) Lactate, (D) VEGF, and (E) GSH level in resected tumor of xenograft mice were measured following specified kits instructions ($n$ = 10 per group). (F) Xenografts neovascularization were measured by the Drabkin's reagent kit. Data are shown as mean ± SD, *($p \leq 0.05$), **($p \leq 0.01$), or ***($p \leq 0.001$). GSH, glutathione; HMGCR, 3-hydroxy-3-methylglutaryl coenzyme A reductase; Lov, lovastatin; ns, not statistically significant; SK, Shikonin; VEGF, vascular endothelial growth factor.

MVA pathway [27]. However, there are still inconsistencies between observational data assessing the relationship between the use of statins and survival in patients suffering from malignancies. Statins have even shown carcinogenic properties in bladder and lung cancer [21,28]. Moreover, when different types of statins were used in different concentration ranges, previous researchers have reported clear evidence of angiogenesis [29,30]. Notably, statins were found to activate protein kinase Akt in normocholesterolemic animals, thus promoting the production of endothelial nitric oxide (NO) to enhance angiogenesis; consequently, statins have been used to treat stroke and cardiac ischemia-reperfusion [31,32]. As upstream factors, Akt and NO are closely related to the regulation of glycolysis and VEGF. These lines of evidence support our conclusion from another angle. Based on these, we highlighted a novel therapeutic recommendation that should be given special consideration when treating RCC patients with statins.

This study identified HMGCR as a novel regulator of PKM2 in RCC cells for the first time. Our data indicated that the inhibition of HMGCR reduced the attenuation of PKM2 protein and thus maintained its stability. Our studies also confirmed the association between PKM2 and tumor proliferation by using a PKM2 gene knockout and a specific inhibitor (Shikonin). PKM2 was originally identified as a glycolytic enzyme that catalyzes the transfer of phosphate groups from phosphoenolpyruvate (PEP) to ADP. However, the role of PKM2 in the metabolic control of glycolysis in cancer cells has been widely reported [33]. The modulation of PKM2 activity by activators or inhibitors might affect tumor growth in vivo [34,35]. The overexpression of PKM2 has been reported to be ubiquitous in human cancers and shown to be associated with poor overall survival, recurrence-free survival, and disease-free survival in a variety of tumors, including kidney cancer [36]. Several previous studies have shown that elevated levels of PKM2 in the circulating blood of patients with RCC can be considered as part of a confirmatory preoperative evaluation [37,38]. Consistent with these concepts, we identified increased levels of PKM2 protein in accelerated RCC tumor tissues in vivo, accompanied by an intensified acidic tumor environment and abnormal vascular proliferation. PKM2 plays a role in various cellular procedures. In addition to controlling glucose metabolism, PKM2 also regulates hypoxia-inducible factor-1α (HIF-1α), β-catenin (β-cat), epidermal growth factor (EGFR), signal transductors, transcriptional activators 3 (STAT3), and other cancer-related factors, thereby promoting cell growth and proliferation [39–41]. Interestingly, PKM2 can also enhance tumor angiogenesis by activating nuclear factor-κB (NF-κB) and HIF-1α and by triggering the secretion of VEGF-A [18], although we did not detect any significant change in HIF-1α at any time point in either our in vivo or in vitro experiments. However, PKM2 promoted tumor VEGF levels; this was confirmed by our animal experiments. Overall, we believe that PKM2 contributed to the progression of RCC and was consistent with the regulation of HMGCR. Therefore, we suggest that PKM2 is a potential tumor marker for monitoring RCC therapy besides HMGCR.

The rapid expression of heat shock protein (HSP) is part of the metabolic stress response of tumors and has proven to be vital for patient survival. This protective mechanism is often referred as the "heat shock response" (HSR). Evidence suggests that many human cancers show the overexpression of HSP, which promotes proliferation, differentiation, immune evasion, invasion, and metastasis of cancer cell [42]. Accumulating evidence also recognizes

HSP90 as a binding partner for PKM2 and has revealed that the expression level of HSP90 is positively related to the abundance of PKM2 protein [19,43]. This notion was consistent with our results that HSP90 could bind with PKM2 and remarkably reduce the level of degradation by the proteasome, thus maintaining the protein stability of PKM2. Although the mechanism responsible for the activation of HSP90 by HMGCR remains unclear, it seems that the loss of isoprene production may be involved; this is because this process can be abolished by the addition of Farnesyl pyrophosphate (FPP) and coenzyme Q10 (COQ10) (S4 Fig). Recent studies indicated that the use of statins significantly increased the expression of Hsp90, while in vitro studies demonstrated that the statin-induced activation of HSP90 was associated with the phosphorylation of endothelial NO synthase (eNOS) [44]. Furthermore, the lack of CoQ10 led to intracellular metabolic stress and inactivation of AKT-related signaling pathways, which directly or indirectly activate HSP90 [45]. In addition, we found that the inhibition of HMGCR increased the protein levels of HSF-1 (S4 Fig). A previous study showed that the administration of atorvastatin in the human coronary artery endothelial cell model up-regulated the levels of HSF-1 protein [46]. HSF-1 might be a key enzyme involved in the mechanism by which HMGCR regulates HSP90. However, the precise identification of the mechanisms by which HMGCR regulates HSP90 has yet to be identified.

The Warburg effect shows that despite having enough oxygen, tumor cells still enhance their glycolysis pathways and use more glucose to synthesize glycolysis intermediates to maintain their cell growth, division, and survival [47]. In this study, we found that HMGCR inhibition did not directly promote the proliferation of cells but increased the rate of glycolysis and enhance the Warburg effect of cells. Moreover, the reprogramming of cellular metabolism provides a microenvironment that is suitable for the accelerated growth of tumors. Aerobic glycolysis by a tumor has a significant interaction with effect on the TME [48]. Enhanced glycolysis and elevated levels of lactate lead to the acidification of the TME, thus activating a variety of low pH-dependent genes that are related to tumor invasion and metastasis [49,50]. TME is the key regulator of carcinogenesis and is composed of many different types of cells, including duo-malignant cancer cells, cancer-associated fibroblasts (CAF), endothelial cells, immune cells, noncancerous cells, and a variety of peptide factors such as growth factors, cytokines, chemokines, and antibodies [51]. The accumulation of lactate in the TME can act as an energy-rich substrate and signal molecule for malignant cancer cells [52]. Acidosis caused by high levels of lactate in the TME can also be beneficial to the metastasis of cancer cells, vascular proliferation, and immunosuppression, thus resulting in a poor clinical prognosis [53]. It has also been shown that high concentrations of lactate found in biopsy tissues from a variety of cancers (kidney, cervical, head and neck, lung, breast, and colorectal cancer) increase the risk of metastasis [54–58]. Moreover, there is a positive correlation between the grade of a cancerous lesion and the concentration of lactate [59]. In addition, as a major component of the TME, CAFs can be activated during tumorigenesis and undergo both morphological and functional changes. CAFs produce large amounts of lactate and releases it into the TME. Adjacent cancer cells then absorb this lactate as an alternative energy source to promoting the occurrence, development, and metastasis of tumors [60]. Correspondingly, the lactate produced by tumor cells increases the production of α-ketoglutarate (AKG) in mesenchymal stem cells (MSCs) and promotes the differentiation of MSCs into CAF [61]. Furthermore, tumor-mediated lactate flux is related to the widespread epigenome reprogramming during the formation of CAF [61]. In our study, the concentration of lactate in the tumor cells and tissues of the HMGCR inhibition group were significantly increased; this was positively correlated with the development of transplanted tumors in mice. Furthermore, CD31 immunohistochemistry indicated an increase in abnormal angiogenesis tumors from the HMGCR inhibition group. These data confirm the conclusions of previous studies.

In that the down-regulation of the glycolysis pathway is a prospective approach with which to inhibit tumor progression and regulate the TME and immune responses [62]. Since HMGCR inhibition exacerbates the Warburg effect of RCC cells, we inferred that this metabolic phenotype would make tumor cells more sensitive to antitumor drugs that mediate glycolysis. Therefore, the use of glycolysis regulators that target PKM2 could be a potential treatment option against tumor growth caused by a decline in HMGCR. Shikonin is a potential PKM2 inhibitor [20] and is known to reduce the rate of glycolysis and lactate production in RCC cells. We found that Shikonin restored the increase in lactate production in tumor tissue of xenograft mice experiencing HMGCR inhibition and reduced the growth of RCC tumors. Moreover, Shikonin is known to exert antitumor effects by itself as it can act as an inhibitor of the TMEM16A chloride channel; it can also prevent activation of the NF-κB pathway and inhibit tumor necrosis factor-α (TNF-α) [63–65]. Based on these facts, we propose that Shikonin could be considered as a potential clinical medication for the treatment or prevention of tumor deterioration caused by HMGCR or glycolysis abnormalities in the clinic.

## Materials and methods

### Xenograft assay

Four to 6 weeks old female BALB/c nude mice were injected subcutaneously of $1 \times 10^7$ cells resuspended in 100 μl PBS into both hind limbs. Five mice were used as a group, and the experiments were performed in duplicate. Drug intervention began after 7 days. Mice were intragastric administrated lovastatin (5 mg/kg) or together with Shikonin (2 mg/kg) at 2-day intervals. Tumors were measured by digital caliper, and volumes were calculated using the following equation: volume = (width$^2 \times$ length)/2. After the tumor volume was last measured, the mice were euthanized by cervical dislocation, the tumor tissue was excised and weighed. Tumor tissue was immediately frozen in liquid nitrogen for future analysis. The formation of neovessels was indirectly determined by measuring the hemoglobin content using the Drabkin's reagent kit (Sigma-Aldrich, St. Louis, MO, USA) according to the manufacturer's protocol. All animals were maintained in a temperature-controlled environment with free access to food and water. All procedures were carried out with the approval of the Animal Ethics Committee of Tsinghua University (Ethics approval number: SCXK2018-0002).

### Reagents and antibodies

Lovastatin, fluvastatin sodium, simvastatin, atorvastatin, CoQ10, and mevalonic acid (MVA) were purchased from Cayman Chemical (Ann Arbor, MI, USA). 2-Deoxy-D-glucose (2-DG), carbonyl cyanide 4-(trifluoromethoxy) phenylhydrazone (FCCP), and oligomycin A were purchased from MedChem Express (Monmouth Junction, NJ, USA). Cholesterol-Water Soluble, FPP, geranylgeranyl pyrophosphate (GGPP), trichloroacetic acid (TCA), CHX, MG132, DNase-free RNase A, triton X-100, crystal violet, and sulforhodamine B (SRB) were obtained from Sigma Aldrich (St. Louis, MO, USA). Antibody sources were as follows: HMGCR (BD Biosciences, San Jose, CA, USA); PKM2 (Cell Signaling, Danvers, MA, USA); Hsp90 (Abcam, Cambridge, Cambridgeshire, UK), β-actin (Sigma-Aldrich, St. Louis, MO, USA); horseradish peroxidase-conjugated secondary antibodies (Jackson Laboratory, Bar Harbor, ME, USA).

### Cell lines and culture

All cell lines were obtained from Cell Bank of Chinese Academy of Science (Shanghai, China). Human renal adenocarcinoma cell lines ACHN and 786-O were cultured in Eagle's Minimum Essential Medium and RPMI-1640 medium (Sigma Aldrich). All culture mediums were

supplemented with 10% fetal bovine serum (FBS), 100 units/mL penicillin G, and 100 g/mL streptomycin (all from Gibco, Thermo Fisher Scientific, Waltham, MA, USA). All cell lines were cultured at 37°C, 5% $CO_2$, and 95% $O_2$ in a humidified atmosphere.

## In vitro cell proliferation assay

The antiproliferative effect of HMG-CoA reductase inhibitors on RCC cells was evaluated by SRB colorimetry. ACHN or 786–0 cells were seeded into a 96-well plate at a density of 5,000 cells per well in a volume of 100 μl/well. Incubate overnight at 37°C in a 5% $CO_2$ incubator. Add 100 μl of lovastatin-containing medium for indicated time drug treatment. Cells attached to the surface of the 96-well plate were fixed with 50 μL of cold 50% TCA at 4°C for 1 hour. After that, the fixing solution was discarded, washed with pure water 5 times, and dried at room temperature. After the 96-well plate was completely dried, 100 μL of 0.4% sulforhodamine B was added for staining. A volume of 200 μl of 10 mM Tris alkaline solution (pH 10.5) was used to dissolve the protein-bound dye, the absorbance at 515 nm was measured using a BioTek Synergy HT plate reader. The cell growth rate is calculated as: relative growth rate (%) = OD (treated) / OD (control).

## Glucose and lactate concentrations measurement

RCC cells were seeded in a 6-well plate ($1 \times 10^6$/well) and incubated at 37°C for 24 hours under indicated treatment. The culture medium was collected for following testing. Glucose and lactate levels were determined using the Glucose (GO) Assay Kit (Sigma) and the Lactate Assay Kit (Cayman Chemical), respectively. Intracellular lactate level was determined by LC–MS system (Thermo Fisher TSQ). For testing the excised tumor, the tumor tissue from each animal were homogenized and suspended (50 mg/ml) in ice-cold buffer containing 10 mM Tris HCl (pH 7.5), 1 mM EDTA, 10 μg/ml leupeptin, 11.5% sucrose, 10 μg/ml pepstatin A, 10 μg/ml aprotinin. The total protein concentration was measured by NanoDrop 2000 (Thermo Scientific). The final glucose and lactate values were normalized to the total protein concentration.

## VEGF and GSH level assessment

The method of processing cells and tumor tissues was as described above. The VEGF protein levels were measured according to the manufacturer's instructions by using an enzyme-linked immunosorbent assay kit (ELISA, R&D Systems, Minneapolis, MN, USA). GSH was determined by using Glutathione Assay Kit (Abcam, ab65322). Final VEGF and GSH levels were normalized to the total protein concentration.

## 2-NBDG uptake assay

After indicated treatments of RCC cells, old medium was exchanged with a glucose-free medium, and subsequently, cells were deprived of glucose for 2 to 3 hours. Then, the medium was changed to a fresh medium containing fluorescent 2-NBDG (Cayman Chemical), and the cells were incubated at 37°C for another 30 minutes. Cells were washed twice with PBS and then analyzed for 2-NBDG uptake by flow cytometry.

## Immunohistochemical analysis

Tumor specimens were fixed in formalin overnight, embedded in paraffin, and cut into 4 mm thick series sections. For immunoelectron labeling, the tissue sections were performed by incubating with CD31 antibodies overnight at 4°C. The sections were then reacted with

appropriate secondary antibodies. The entire section was scanned by a scanning system that consists of fluorescence microscope (Xcellence, Olympus, Tokyo, Japan), Image-Pro Plus 6.0 (Media Cybernetics, Silver Spring, Maryland), and SPSS V19.0.

## Co-immunoprecipitation (Co-IP) assay

Cells were lysed using NP-40 and total protein lysates were extracted. Mixed 500 μg of total protein with 1 μg primary antibody or IgG and put on a shaker at 4˚C for 4 h. Protein G beads (GE Healthcare Life Sciences, Boston, MA, USA) were subsequently added into the mixture and incubated overnight at 4˚C with shaking. The beads were then washed 3 times with IP buffer. Mixed the 5X sample loading buffer with the beads and boiled for 10 minutes. Final products were subjected to western blot analysis.

## Glutathione S transferase (GST) pull-down assay

Cells were lysed in a lysis buffer containing a protease inhibitor (Roche, Shanghai, China). Cell harvested was then incubated with GST-HSP90 or control GST on a shaker at 4˚C for 6 h to promote protein–protein interactions. The suspension was further incubated with Glutathione beads (Sigma-Aldrich) for 2 hours. The protein-bead complex was then collected by centrifugation at 1,000$g$ for 5 minutes. After washing the beads 3 times with washing buffer, the protein was eluted. Finally, the eluted products were analyzed by western blot.

## Seahorse extracellular flux analyses

ACHN or 786–0 cells were seeded at 8,000 cells/well into XFp cell culture plates and incubated overnight. The next day, following the procedure recommended in the instrument manual, wash the cells and change to XF assay medium, and incubate for 1 hour at 37˚C in a CO2-free incubator. The XFp extracellular flux analyzer (Seahorse Bioscience, North Billerica, MA, USA) was then used to determine glycolysis flux and cellular respiration. The indicated cell stress agents were injected at preset time points, and the ECAR and oxygen consumption rate (OCR) were recorded.

## Western blotting

Cells were lysed in a RIPA buffer containing a protease inhibitor (Roche) and a phosphatase inhibitor (Biochemica, Darien, IL, USA). After total protein was obtained, it was diluted in SDS-PAGE protein sample buffer and boiled at 95˚C for 5 minutes. A total of 20 μg of each protein were resolved by SDS-PAGE. Separated proteins were transferred to Immobilon-P PVDF Transfer Membranes (Millipore, Temecula, CA, USA) at 300 mA. The membrane was blocked with 5% BCA (bicinchoninic acid) for 1 hour at room temperature, and then incubated with the designated primary antibody at 4˚C overnight. Next day, incubate the blot with a suitable horseradish peroxidase-conjugated secondary antibody for 45 minutes at room temperature. The blots were then visualized with SuperSignal Chemiluminescent Substrate (Thermo Scientific) and detected by chemiluminescence imaging system (Perkin Elmer, Waltham, MA, USA).

## Quantitative real-time PCR

Quantitative PCR was performed using cDNA prepared from 1 μg of RNA of treated cells, with SYBR Green (Molecular Probes, San Jose, CA, USA) in the StepOnePlus Real-Time PCR System (Applied Biosystems, Foster City, CA, USA), together with primers synthesized using our own designed templates. Primer sequences were designed employing Thermo Fisher

Scientific's online Designer software and then verified with NCBI's Primer–BLAST software to confirm specific recognition of the target genes. Gene expression levels were normalized to the eukaryotic translation initiation factor (TIF) expression. Three independent experiments with 3 replicates per group were analyzed for each primer. All the data were statistically analyzed by unpaired *t* test.

### cDNA expression and mRNA knockdown in vitro

The *sh*RNA sequences that specifically target HSP90, PKM2, and HMGCR were cloned into the lentiviral vector pLKO.1 plasmid (Addgene, 10,878) and verified by sequencing.

 *sh*HSP90-1:5′-CCGGGTTATCCTACACCTGAAAGAACTCGAGTTCTTTCAGGTGTAG GATAACTTTTTG-3′,

 *sh*HSP90–2:5′-CCGGTACTTGGAGGAACGAAGAATACTCGAGTATTCTTCGTTCCT CCAAGTATTTTTG-3′,

 *sh*PKM2–1:5′-CCGGGTTCGGAGGTTTGATGAAATCCTCGAGGATTTCATCAAACC TCCGAACTTTTTTG-3′,

 *sh*PKM2-2:5′-CCGGGCCCGAGGCTTCTTCAAGAAGCTCGAGCTTCTTGAAGAAGC CTCGGGCTTTTTTG-3′,

 *sh*HMGCR-1:5′-CCGGCTATGATTGAGGTCAACATTACTCGAGTAATGTTGACCTC AATCATAGTTTTTG-3′,

 *sh*HMGCR-2:5′-CCGGGGGTTCTAAAGGACTAACATAACTCGAGTTATGTTAGTCCT TTAGAACCTTTTTG-3′.

 The empty plasmid pLenti-CMV and the plasmid containing PKM2 (PKM) cDNA (accession number NM_182470), HSP90 cDNA (accession number NM_007355) were obtained from Public Protein/Plasmid Library. Lentiviruses were established by transfecting the lentiviral vectors and packaging with the MISSION packing mix (Sigma) into human embryonic kidney cell HEK293T cells following the manufacturer's protocol. ACHN and 786-O cells were stably transduced with lentivirus for cDNA expression or mRNA knockdown and selected with puromycin for 7 days before further analysis.

### Statistical analyses

Data expressed as the mean ± SD as indicated. The Microsoft Excel and GraphPad Prism 5 software were used for the statistical analysis including two-tailed distribution Student *t* test, two-way ANOVA, or Tukey test. Significant differences were indicated as $^*(p \leq 0.05)$, $^{**}(p \leq 0.01)$, or $^{***}(p \leq 0.001)$ according to the level of significance.

## Supporting information

**S1 Fig. HMGCR inhibition promotes the proliferation of RCC cells through regulating lactate production and glucose consumption.** (A) Dose-dependent lovastatin treated on the proliferation of ACHN and 786-O cells. (B) Cell proliferation analysis in ACHN and 786-O cells treated with atorvastatin (Atr), simvastatin (Sim), or fluvastatin (Flu). (C) Western blotting analysis of HMGCR after lentiviral knockdown as exemplified in ACHN and 786-O cells. (D) Kinetic ECAR response of HMGCR knockdown or lovastatin-treated 786-O cells. (E) Glucose consumption of ACHN or 786-O cells under 24-hour lovastatin treatment. (F) Intracellular 2-NBDG glucose uptake in lovastatin-treated cells was analyzed by flow cytometry. The data are represented as the mean ± SD from 3 independent experiments, $^*(p \leq 0.05)$, $^{**}(p \leq 0.01)$, or $^{***}(p \leq 0.001)$. 2DG, 2-deoxy-D-glucose; Atr, atorvastatin; Ctrl, control; ECAR, extracellular acidification rate; Flu, fluvastatin; Glu, glucose; HMGCR, 3-hydroxy-3-methylglutaryl coenzyme A reductase; Lov, lovastatin; NS, non-specific; Oligo, oligomycin;

RCC, renal cell carcinoma; Sim, simvastatin; *wt*, wild type.
(TIF)

**S2 Fig. PKM2 is the key regulator of HMGCR-mediated glycolysis elevation.** (A) The effect of lovastatin or *sh*HMGCR intervention on the protein levels of glycolysis-related enzymes. (B) Western blotting analysis of PKM2 after lentiviral knockdown as exemplified in ACHN cells. (C) Glucose consumption in PKM2 knockdown ACHN cells with or without lovastatin treatment. (D) Western blotting analysis of PKM2 with exogenous overexpression as exemplified in ACHN cells. (E) Glucose consumption in PKM2 overexpression ACHN cells. The data are represented as the mean ± SD from 3 independent experiments, $^*(p \leq 0.05)$, $^{**}(p \leq 0.01)$, or $^{***}(p \leq 0.001)$. Glut1, glucose transporter 1; HIF-1α, hypoxia-inducible factor-1α; HMGCR, 3-hydroxy-3-methylglutaryl coenzyme A reductase; LDHA, lactate dehydrogenase A; Lov, lovastatin; NS, non-specific; PKM2, pyruvate kinase M2.
(TIF)

**S3 Fig. HMGCR inhibition did not alter *pkm2* expression.** *pkm2* expression level in lovastatin-treated (left panel) or HMGCR knockdown (right panel) ACHN cells. The data are represented as the mean ± SD from 3 independent experiments. HMGCR, 3-hydroxy-3-methylglutaryl coenzyme A reductase; Lov, lovastatin; NS, non-specific; *pkm2*, pyruvate kinase M2.
(TIF)

**S4 Fig. HMGCR inhibition induces the up-regulation of HSP90.** (A) Western blotting analysis of HSP90 after lentiviral knockdown as exemplified in ACHN cells. (B) Supplemental effect of MVA pathway downstream metabolites on HSP90 level in lovastatin-treated ACHN cells. (C) Lovastatin and *sh*HMGCR intervention increases the level of HSF-1 protein, and the effect is rescued when mevalonate is supplemented. (D) Western blotting analysis of HSP90 with exogenous overexpression as exemplified in ACHN cells. Chol, cholesterol; CoQ10, coenzyme Q10; Ctrl, control; FPP, farnesyl pyrophosphate; GGPP, geranylgeranyl pyrophosphate; HMGCR, 3-hydroxy-3-methylglutaryl coenzyme A reductase; HSF-1, heat shock transcription factor-1; HSP90, heat shock protein 90; Lov, lovastatin; MVA, mevalonate; NS, non-specific.
(TIF)

**S5 Fig. Inhibitory effect of PKM2 inhibitor in HMGCR inhibition RCC tumor.** (A) Immunohistochemical staining of CD31+ in ACHN cell xenografts (scale bar: 50 μm). (B) Extracellular lactate level, (C) glucose consumption, and (D) ECAR rate of ACHN cell with or without lovastatin and Shikonin treatment. The data are represented as the mean ± SD from 3 independent experiments, $^*(p \leq 0.05)$, $^{**}(p \leq 0.01)$ or $^{***}p \leq 0.001$. 2DG, 2-deoxy-D-glucose; ECAR, extracellular acidification rate; Glu, glucose; HMGCR, 3-hydroxy-3-methylglutaryl coenzyme A reductase; Lov, lovastatin; Oligo, oligomycin; PKM2, pyruvate kinase M2; RCC, renal cell carcinoma; SK, Shikonin; *wt*, wild type.
(TIF)

**S1 Raw Images.**
(PDF)

## Author Contributions

**Conceptualization:** Jiajun Huang.

**Data curation:** Jiajun Huang, Xiaoyu Zhao, Xiang Li.

**Formal analysis:** Jiajun Huang, Xiaoyu Zhao.

**Funding acquisition:** Jiajun Huang, Shengli Mi.

**Investigation:** Xiang Li.

**Methodology:** Jiajun Huang.

**Project administration:** Shengli Mi.

**Supervision:** Jiajun Huang, Shengli Mi.

**Validation:** Jiwei Peng, Weihao Yang.

**Writing – original draft:** Jiajun Huang, Xiaoyu Zhao.

**Writing – review & editing:** Jiajun Huang, Xiaoyu Zhao.

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
