## [Editor Report · Decision Letter 0]

15 Jun 2020

Dear Dr Huang, 

Thank you for submitting your manuscript entitled "Regulation of 3-Hydroxy-3-methylglutaryl coenzyme A reductase on glycolysis in renal cell carcinoma" for consideration as a Research Article by PLOS Biology.

Your manuscript has now been evaluated by the PLOS Biology editorial staff as well as by an academic editor with relevant expertise and I am writing to let you know that we would like to send your submission out for external peer review.

Please re-submit your manuscript within two working days, i.e. by Jun 17 2020 11:59PM.

Kind regards,

Ines

--

Ines Alvarez-Garcia, PhD

Senior Editor

PLOS Biology

Carlyle House, Carlyle Road

Cambridge, CB4 3DN

+44 1223–442810

---

## [Decision Letter · Decision Letter 1]

5 Oct 2020

Dear Dr Huang,

Thank you very much for submitting your manuscript "Regulation of 3-Hydroxy-3-methylglutaryl coenzyme A reductase on glycolysis in renal cell carcinoma" for consideration as a Research Article at PLOS Biology. Thank you also for your patience as we completed our editorial process, and please accept my sincere apologies for the long delay in providing you with our decision. Your manuscript has been evaluated by the PLOS Biology editors, an Academic Editor with relevant expertise, and by two independent reviewers.

The reviews are attached below. You will see that the reviewers agree that the results are novel and interesting for the field, but they also raise several concerns that would need to be addressed to consider the manuscript for publication. Reviewer 1 would like you to explore HMGCR function further, whereas Reviewer 2 asks for several controls that need to be performed to confirm the conclusions. In addition, this reviewer would like you to add some stats, improve the title and the text and to clarify some points.

In light of the reviews, we will not be able to accept the current version of the manuscript, but we would welcome re-submission of a revised version that takes into account the reviewers' comments. We cannot make any decision about publication until we have seen the revised manuscript and your response to the reviewers' comments. Your revised manuscript is also likely to be sent for further evaluation by the reviewers.

We expect to receive your revised manuscript within 3 months. 

**IMPORTANT - SUBMITTING YOUR REVISION**

*Re-submission Checklist*

*Published Peer Review*

*PLOS Data Policy*

*Blot and Gel Data Policy*

Best wishes,

Ines

--

Ines Alvarez-Garcia, PhD,

Senior Editor,

ialvarez-garcia@plos.org,

PLOS Biology

Reviewers’ comments

Rev. 1:

This is an interesting study that reports a novel mechanism underlying inhibition of HMCGR accelerating RCC progression via the HSP90/PKM2 axis. I have some comments, questions, and suggestions for the authors to consider which may serve to further strengthen the manuscript:

1. HMGCR inhibition did not directly impact the proliferation of RCC cells in vitro, but promotes tumor progression in vivo. Previous studies suggest that cancer cell-produced lactate activates CAFs or TAMs, which in turn promote tumor progression. Please discuss this.

2. Does HMGCR inhibition affect other glycolysis-related regulators? Such as HK2, GLUT1, and LDHA. Please clarify this.

3. Does HMGCR function as a transcription factor? If no, how HMGCR regulate HSP90 transcription in RCC cells? Please explore the potential mechanism.

Rev. 2:

The manuscript submitted by Huang et al. ("Regulation of 3-Hydroxy-3-methylglutaryl coenzyme A reductase on glycolysis in renal cell carcinoma") describes the role of HMGCR inhibition in promoting tumor growth of renal cell carcinoma. The authors show that loss of HMGCR function by treatment with lovastatin or shRNA upregulated the glycolytic enzyme PKM2, through stabilization by HSP90. Lovastatin treatment increased xenograft tumor growth, but PKM2 inhibition alone or in combination with lovastatin, showed no difference in tumor size. The impact of the mevalonate pathway and statin treatment on cancer development is of great interested, particularly as conflicting reports have demonstrated both tumor promoting and inhibiting roles. The authors use various biochemical assays to convincingly show a relationship between HMGCR inhibition and HSP90-mediated PKM2 regulation in RCC cell lines; however, inclusion of proper controls in many cases makes the current paper unacceptable for publication. It is also very poorly put together and harbors numerous grammatical and typographical errors.

Major Concerns:

1) The manuscript is poorly written with many grammatical and spelling errors that make the authors' interpretation of the data very difficult to understand. The authors need to extensively revise the writing before the manuscript can be considered for publication anywhere.

2) The current title does not accurately describe the work presented in the manuscript, as very few experiments actually explore the regulation of HMGCR and glycolysis (only lactate production and ECAR measurements). A more appropriate title might be something like: "HMGCR inhibition stabilizes glycolytic enzyme PKM2 through HSP90 upregulation to support tumor growth of renal cell carcinoma"

3) Statistical analysis of tumor volume and weight are missing in Figures 1A and 1B.

4) For all usage of shRNAs, please include evidence of knockdown.

5) In Figure 2D, treatment with lovastatin decreases ECAR while shHMGCR increases ECAR. However, in the results section, this data is described as "ECAR…increased with HMGCR inhibition or knockdown". Can the authors clarify this discrepancy?

6) The authors claim to have "examined the expression levels of glycolysis enzymes and already known upstream regulators". Other than levels of PKM2 in Figure 3, have the expression levels of other glycolytic enzymes upon inhibition of HMGCR been investigated? A previous study by the authors identified HMGCR regulation of lactate in skeletal muscle through enolase modulation (Huang et al. EBioMedicine 2019). Does this hold true in RCC?

7) Figure 3F and Supp Figure 2A are described as "a decrease in the expression of PKM2 protein level resulted in lactate production and glucose consumption decline, and this effect could not be reversed by further application of lovastatin". In order to make this claim, a control of shPKM2 alone must be included in this experiment. Otherwise, the results show that loss of PKM2 completely reverses that induction of lactate production and glucose consumption induced by lovastatin.

8) Related to comment 6- in the writing, the authors mistakenly reference Figure 3E.

9) Also related to comment 6- what are the comparisons made for the indicated statistics?

10) "…exogenous expression of PKM2 significantly increased lactate production.." should reference Figure 3G, not 3H.

11) "consistent with the trend…" describing Figure 3I seems out of place. Please move this description to before the introduction of the CHX experiments.

12) Figure 3K- "… MG132 to rescue the decline of PKM2 induced by mevalonate supplementation." In the displayed experiment, the authors have not supplemented with mevalonate. Inclusion of controls including lov alone, shHMGCR alone, and rescues with MVA would greatly strengthen this data.

13) "As shown in Figure 5C…" references Figure 4C.

14) Figure 4C: change label to IP: PKM2.

15) Figure 5B: lanes need to be labeled.

16) Figure 5F: Data is not consistent with the rest of Figure 5 or the description of the figure. The +shHSP90 lane has reduced ubiquitination of PKM2. The authors need to include a control blot of HSP90 to confirm loss of HSP90 in this experiment.

17) Figure 6: This xenograft experiment should be repeated including untreated tumors (control shRNA, saline only), as well as Lovastatin alone in order to show that any phenotypes are truly "reversed". Additionally, inclusion of shHMGCR tumors with Shikonin treatment would provide additional support to the translational claims made by the authors.

18) The authors mention the use of HMGCR as a potential biomarker for treatment with a glycolysis inhibitor. What is the baseline expression of HMGCR, HSP90, and PKM2 in the cell lines used for the experiments? Would inhibition of PKM2, perhaps through Shikonin treatment, be more effective in tumors with the lowest levels of HMGCR?

19) What are the levels of HMGCR in primary tumor samples compared to normal kidney tissue, based on TCGA datasets?

---

## [Decision Letter · Decision Letter 2]

17 Feb 2021

Dear Dr Huang,

Thank you for submitting your revised Research Article entitled "HMGCR inhibition stabilizes the glycolytic enzyme PKM2 to support the growth of renal cell carcinoma by upregulating HSP90" for publication in PLOS Biology. I have now obtained advice from the two original reviewers and have discussed their comments with the Academic Editor. 

Based on the reviews (attached below), we will probably accept this manuscript for publication, provided you satisfactorily address the remaining point raised by Reviewer 2. Please also make sure to address the following data and other policy-related requests. In addition, we would like to suggest a change in the title for improvement:

"HMGCR inhibition stabilizes the glycolytic enzyme PKM2 to support the growth of renal cell carcinoma"

or

"HMGCR inhibition leads to Hsp90-dependent stabilization of the glycolytic enzyme PKM2 to support the growth of renal cell carcinoma"

We expect to receive your revised manuscript within two weeks. 

*Published Peer Review History*

*Early Version*

Sincerely,

Ines

--

Ines Alvarez-Garcia, PhD,

Senior Editor,

PLOS Biology

ETHICS STATEMENT:

-- Thank you for including the ethics statement in the Methods section. Please include the license and approval number.

DATA POLICY:

I am aware you have submitted some files in pzf files, but we cannot access them at the moment. Nevertheless, we want to make sure you have provided all the individual numerical values that underlie the summary data displayed in the following figure panels as they are essential for readers to assess your analysis and to reproduce it:

Fig. 1A-C; Fig. 2A-E; Fig. 3F-I, J; Fig. 4D-F; Fig. 5F; Fig. 6A-F; Fig. S1A, B, D-F; Fig. S2C, E; Fig. S3 and Fig. S5B-D

While you can provided in the format you wish, as far as the set is complete, most of our authors provide an excel file containing multiple sheets in one excell file with labeled panels of a single or even several figures and saved using exactly the following convention: S1_Data.xlsx (using an underscore).

In the acknowledgements, you state that: “All data generated or analyzed in this study are included either in this article and/or in the supplementary information files. Additional data for this paper may be requested from the corresponding author: Jiajun Huang (huangjiajun0401@sz.tsinghua.edu.cn) or Shengli Mi (mi.shengli@sz.tsinghua.edu.cn)." If all the data is included in the manuscript, please delete the second sentence, as it is confusing for readers. In addition, this should be the Data Availability Statement, rather than Acknowledgements.

For figures containing Flow Cytometry data, we ask that you provide FCS files and a picture showing the successive plots and gates that were applied to the FCS files to generate the figure.

Reviewers' comments 

Rev. 1:

The authors have responded to all my comments.

Rev. 2:

The authors responded to my concerns well, with one minor exception: the authors claim that they have examined the expression levels of glycolysis enzymes and known upstream regulators (Comment #6) is previous review. They have now assesses a number of upstream regulators (e.g. TIGAR, c-Myc), but still not other enzymes beyond PKM2. The minor edit is that this should be made clear in the text.

---

## [Editor Report · Decision Letter 3]

17 Mar 2021

Dear Dr Huang,

Thank you for submitting your revised Research Article entitled "HMGCR inhibition stabilizes the glycolytic enzyme PKM2 to support the growth of renal cell carcinoma" for publication in PLOS Biology.

We are almost satisfied with the manuscript, but we need you to address the following remaining points:

1. Please add the license/approval number to the Ethics statement.

2. For figures containing Flow Cytometry data, we ask that you provide the FCS files. We would suggest you deposit them in the FlowRepository (http://flowrepository.org/). And please make sure they are made publicly available.

3. *Blurb*

Please also provide a blurb which will be included in our weekly and monthly Electronic Table of Contents, sent out to readers of PLOS Biology, and may be used to promote your article in social media. The blurb should be about 30-40 words long and is subject to editorial changes. It should, without exaggeration, entice people to read your manuscript. It should not be redundant with the title and should not contain acronyms or abbreviations. For examples, view our author guidelines: https://journals.plos.org/plosbiology/s/revising-your-manuscript#loc-blurb

We expect to receive your revised manuscript within one week. 

-  a cover letter that should detail your responses to any editorial requests.

Sincerely,

Ines

--

Ines Alvarez-Garcia, PhD,

Senior Editor,

PLOS Biology

---

## [Editor Report · Decision Letter 4]

19 Mar 2021

Dear Dr Huang,

On behalf of my colleagues and the Academic Editor, Heather Christofk, I am pleased to say that we can in principle offer to publish your Research Article entitled "HMGCR inhibition stabilizes the glycolytic enzyme PKM2 to support the growth of renal cell carcinoma" in PLOS Biology, provided you address any remaining formatting and reporting issues. These will be detailed in an email that will follow this letter and that you will usually receive within 2-3 business days, during which time no action is required from you. Please note that we will not be able to formally accept your manuscript and schedule it for publication until you have made the required changes.

PRESS

Thank you again for supporting Open Access publishing. We look forward to publishing your paper in PLOS Biology. 

Sincerely, 

Ines

--

Ines Alvarez-Garcia, PhD 

Senior Editor 

PLOS Biology